# Synthesis of FePcS–PMA–LDH Cointercalation Composite with Enhanced Visible Light Photo-Fenton Catalytic Activity for BPA Degradation at Circumneutral pH

**DOI:** 10.3390/ma13081951

**Published:** 2020-04-21

**Authors:** Fenglian Huang, Shiqiang Tian, Yan Qi, Erping Li, Liangliang Zhou, Yaqun Qiu

**Affiliations:** 1Hunan Provincial Key Laboratory of Water Pollution Control Technology, Hunan Academy of Environmental Protection Sciences, Changsha 410004, Hunan, China; 2hfl@163.com (F.H.); hnhkytsq@sina.com (S.T.); qiyan887@126.com (Y.Q.); lvtu1214@sina.com (E.L.); zhousunset@foxmail.com (L.Z.); 2College of Environmental Science and Engineering, Central South University of Forestry and Technology, Changsha 410004, Hunan, China

**Keywords:** photo-Fenton, iuron tetrasulfophthalocyanine, phosphomolybdic acid, layered double hydroxides, circumneutral pH

## Abstract

(1) Background: Iron tetrasulfophthalocyanine with a large nonlinear optical coefficient, good stability, and high catalytic activity has aroused the attention of researchers in the field of photo-Fenton reaction. Further improvement of the visible light photo-Fenton catalytic activity under circumneutral pH conditions for their practical application is still of great importance. (2) Methods: In this paper, iron tetrasulfophthalocyanine (FePcS) and phosphomolybdic acid (PMA) cointercalated layered double hydroxides (LDH) were synthesized by the ion-exchange method. All samples were fully characterized by various techniques and the results showed that FePcS and PMA were successfully intercalated in layered double hydroxides and the resulted compound exhibited strong absorption in the visible light region. The cointercalation compound was tested as a heterogeneous catalyst for the visible light photo-Fenton degradation of bisphenol A (BPA) at circumneutral pH. (3) Results: The results showed that the degradation and total organic carbon removal efficiencies of bisphenol A were 100% and 69.2%, respectively. (4) Conclusions: The cyclic voltammetry and electrochemical impedance spectroscopy measurements demonstrated that the main contribution of PMA to the enhanced photo-Fenton activity of FePcS–PMA–LDH comes from the acceleration of electron transfer in the reaction system. Additionally, the possible reaction mechanism in the photo-Fenton system catalyzed by FePcS–PMA–LDH was also proposed.

## 1. Introduction

In recent years, great endeavors have been made to develop efficient chelating agents for stabilizing iron and enhancing the photo-Fenton degradation of refractory organic contaminants in water under neutral pH conditions [1,2,3,4]. To date, a large number of Fe complexes such as Fe–ethylenediaminetetraacetic acid, Fe–oxalate, Fe–ethylenediamine–*N*, *N*′–disuccinic acid, and iron phthalocyanine complex (FePc) have been reported as photo-Fenton catalysts [5,6,7,8], in which FePc has been found to have good response to visible light [7,9,10,11]. Under visible light irradiation, effective energy transfer can occur between the excited FePc and H_2_O_2_, resulting in the formation of the •OH radical, which can oxidize organic matter with high efficiency [12]. However, the practical application of FePc is limited because of the low quantum yield of •OH resulting from the short life of excited MPc [13].

Coupling FePc with polyoxometalates (POM) may be an effective strategy to improve the utilization of electronically excited FePc. Previous studies have shown that POM can act as an electron acceptor and lead to the formation of reduced POM [14,15,16,17,18], which can not only accelerate the transfer of Fe(III) to Fe(II) in FePc, but can also activate hydrogen peroxide or oxygen to produce various reactive oxygen species with strong oxidizing property [19]. Inspired by these studies, we expect that the introduction of phosphomolybdic acid (PMA), a Keggin polyoxometalate, can greatly enhance the photo-Fenton catalytic activity of FePc. Considering the high solubility of FePcS and PMA in aqueous media, the immobilization of FePcS–PMA is of great importance for the recovery and reuse of this catalyst. Layered double hydroxides (LDHs) have been considered as attractive materials for their high surface area, variable gallery height, and strong adsorption performance [20,21,22]. Moreover, due to the remarkable anion-exchange capacity, LDHs are very suitable supporters for the immobilization of the anionic complex [23,24].

With this understanding, we report herein the synthesis of FePcS–PMA–LDH composite catalyst using ZnAl–layered double hydroxide (ZnAl–LDH) as the host material to support FePcS and PMA. The as-prepared catalyst was used as an efficient heterogeneous catalyst for the photo-Fenton degradation of bisphenol A (BPA) under visible light irradiation and neutral pH.

## 2. Materials and Methods

### 2.1. Synthesis of FePcS–PMA–LDH

ZnAl–LDH with a Zn/Al ratio of 2 was prepared through the co-precipitation method at a constant pH of 6. FePcS was synthesized and purified according to the method of Griffin [25]. To synthesize the FePcS–PMA–LDH, a 100 mL suspension containing 2.0 g ZnAl–LDH was added dropwise to a 50 mL mixed solution of FePcS/PMA (0.02 g:0.06 g) under magnetic stirring and an inert atmosphere. In addition, 20 mL of ethylene glycol and 20 mL of ethanol were added to the reacting mixture, which was sonicated for 10 min to obtain a homogeneous dispersion and was then magnetically stirred for 24 h. The obtained precipitate was filtered and washed with deionized water under an inert atmosphere, and then dried at 60 °C under vacuum.

### 2.2. Characterization

The x-ray diffraction (XRD) patterns were investigated by a LabX-6000 diffractometer (Shimadzu, Hong Kong, China). The Fourier transform infrared (FTIR) spectra were recorded by a Nicolet 380 FTIR spectrometer using KBr pellets at room temperature. The zeta potential values were measured using a Zeta sizer (nano zs90, Malvern Instruments, Worcestershire, UK). The Brunauer–Emmett–Teller (BET) measurements of the materials were taken on a IGA 100B instrument (Hiden, Warrington, England). Diffuse reflectance–UV–Vis (DRS–UV–Vis) measurements were performed on a Shimadzu UV-2550 double-beam digital spectrometer (Hong Kong, China) equipped with conventional components of a reflectance spectrometer. Scanning electron microscopy (SEM) images were obtained using a scanning electron microscope (SEM, JSM-6360LV, JEOL, Peabody, MA, USA).

The electrochemical measurements were performed with a CHI660 Electrochemical Workstation (Shanghai Chenhua Instrumental Co. Ltd., Shanghai, China) and a conventional three-electrode system, in which a ZnAl–LDH/FePcS–LDH/FePcS–PMA–LDH modified glassy carbon electrode (GCE), a platinum plate, and a saturated calomel electrode were used as the working, counter, and reference electrodes, respectively. The electrolyte was 0.1 M KCl solution containing 5 mM K_3_[Fe(CN)_6_]/K_4_[Fe(CN)_6_] (1:1) mixture. Cyclic voltammetry (CV) experiments were conducted at a sweep rate of 100 mV/s and the scan range was from −0.2 to +0.8 V. The electrochemical impedance spectra (EIS) were obtained over the frequency range from 100 kHz to 0.01 Hz with an AC signal amplitude of 10 mV.

### 2.3. Photocatalytic Tests and Analytical Methods

The photocatalytic experiments were conducted in a photochemical reaction instrument (BL-GHX-CH500, Xi’an Depai Biotech. Co. Ltd., Xi’an, China) and a 500 W xenon lamp was applied as the visible-light source. Typically, 40 mg FePcS–PMA–LDH was added to 100 mL of BPA solution with the initial concentration of 10 mg L^−1^. The initial pH value of the solution was adjusted by adding NaOH or HNO_3_ solutions. The mixed solution was magnetically stirred for 30 min in the dark to achieve adsorption equilibrium between FePcS–PMA–LDH and BPA. Then, the xenon lamp was turned on and H_2_O_2_ was added to the BPA solution. At predetermined time intervals, the sample was taken out and centrifuged to obtain the supernatant for analysis. The concentration of the remaining BPA in the aqueous solution was determined with high efficiency liquid chromatography (HPLC, Agilent 1260). The concentration of Total Organic Carbon (TOC) during the degradation solutions was measured by a TOC analyzer (Shimadzu TOC-L CPH CN 200, Kyoto, Japan) equipped with an auto-sampler. Electron spin resonance (ESR) spectra of •OH, O_2_^•−^, and ^1^O_2_ were tested by an ESR spectrometer (JEOL JES-FA, Peabody, MA, USA).

## 3. Results

### 3.1. Characterization of Catalyst

Figure 1 shows the XRD patterns of ZnAl–LDH, FePcS–LDH, and FePcS–PMA–LDH. As demonstrated in the diffractograms of ZnAl–LDH, the diffraction peaks at 2θ = 9.8° (003), 19.8° (006), 34.2° (009) matched well with the characteristic peaks of the reported ZnAl–LDH [26,27,28]. The above peaks can also be found in the patterns of FePcS–LDH and FePcS–PMA–LDH, but slightly shifted to lower angles when compared with ZnAl–LDH. Additionally, the value of the basal distance increased from 0.894 nm for ZnAl–LDH to 1.037 nm and 1.046 nm for FePcS–LDH and FePcS–PMA–LDH, respectively (Table 1). These results indicate the interlayer space of ZnAl–LDH is expanded after intercalated by FePcS and FePcS–PMA.

The molecular structure and chemical nature of ZnAl–LDH, FePcS–LDH, and FePcS–PMA–LDH were analyzed by FTIR spectra, which are shown in Figure 2. In the spectrum of ZnAl–LDH, the characteristic absorption peaks at 3446, 1630, and 1403 cm^−1^ can be attributed to the O–H stretching vibration of interlayer water molecules, the bending vibration of H_2_O in the brucite-like layer, and the stretching vibration of nitrate, respectively [29,30,31]. Compared to the ZnAl–LDH sample, the FTIR spectra of FePcS–LDH and FePcS–PMA–LDH appeared as a new absorption band at 1120 cm^−1^, which was assigned to the stretching vibration absorption of the S=O bond in FePcS [32,33], whereas the infrared absorbance near 647 cm^−1^ was assigned to the vibration absorption of C–H out-of-plane ring bend on the aromatic ring in FePcS [34]. Furthermore, in contrast to ZnAl–LDH, the FTIR spectrum of FePcS–PMA–LDH appeared as characteristic peaks of PMA. The absorption peak at 871 cm^−1^ corresponded to the stretching vibrations of Mo–O_c_–Mo and Mo–O_b_–Mo bands, and the stretching vibrations at 966 cm^−1^ and 1064 cm^−1^ referred to Mo=O_d_ and P–O_a_ bands [35,36,37].The results of the FTIR spectra further confirmed the successful modification of FePcS and FePcS–PMA in ZnAl–LDH.

Figure 3 presents the evolution of the zeta potential as a function of pH for the ZnAl–LDH, FePcS–LDH, and FePcS–PMA–LDH with the particle concentration of 2 g/L. As can be seen, the zeta potential values of ZnAl–LDH were positive in a wide range of pH (pH = 4–10), which resulted from the structural charge and the hydroxyl groups located on the surface of the ZnAl–LDH particles [38]. However, the zeta potential values for FePcS–LDH and FePcS–PMA–LDH were reversed to negative because of the immobilization of FePcS and FePcS–PMA. The physical electrostatic force between the LDH’s structural positive charges and anionic charge compounds is one of the main interaction forces between LDHs and the anionic compounds [39]. Therefore, the immobilized anionic compounds can neutralize the structural charges of ZnAl–LDH. It should be noted that FePcS–PMA exhibited more negative zeta potential values than FePcS–LDH. This result also reflects that FePcS and PMA have been cointercalated into the interlayer galleries of ZnAl–LDH.

The surface morphologies of the catalysts were examined by SEM. As shown in Figure 4, the surface of ZnAl–LDH was smooth, while relatively rough surfaces of FePcS–LDH and FePcS–PMA–LDH particles can be observed, which may result from the homogeneous distributions of FePcS and FePcS–PMA on the LDHs surface. It also should be noted that FePcS–PMA–LDH exhibited a rougher surface morphology compared with FePcS–LDH, indicating that FePcS–PMA–LDH can provide more adsorption sites than FePcS–LDH.

The specific area, pore volume, and pore size of ZnAl–LDH, FePcS–LDH, and FePcS–PMA–LDH were measured by nitrogen adsorption–desorption isotherms. As shown in Figure 5, ZnAl–LDH, FePcS–LDH, and FePcS–PMA–LDH exhibited type H3 hysteresis loops of the Brunauer–Deming–Deming–Teller (BDDT) type Ⅳ isotherm [40], illustrating the above catalysts belong to the characteristic of mesoporous materials [41,42].

The specific areas, pore volumes, and pore sizes of the three samples are listed in Table 2, where the specific area, pore volume, and pore size of ZnAl–LDH were 9 m^2^/g, 0.012 cm^3^/g, and 16.6 nm, respectively. After loaded with FePcS, the values increased to 16 m^2^/g, 0.031 cm^3^/g, and 18.0 nm, respectively. For FePcS–PMA–LDH, the corresponding values (22 m^2^/g, 0.041 cm^3^/g, and 21.7 nm) were higher than those of ZnAl–LDH and FePcS–LDH, suggesting that FePcS–PMA–LDH can provide more catalytic active sites in the process of pollutant degradation [43]. These results are in agreement with the SEM observations.

From the UV–Vis diffused reflectance spectra of ZnAl–LDH, FePcS–LDH, and FePcS–PMA–LDH (Figure 6), it can be seen that FePcS–PMA–LDH had two strong absorption bands at around 650 nm and 350 nm, while ZnAl–LDH exhibited an absorption only in the ultraviolet light region (250–340 nm). The additional UV–Vis absorbance for FePcS–PMA–LDH can be ascribed to the existence of FePcS with the “Q-band” and “B-band” absorption band [44]. Additionally, because of the existence of PMA, the region of absorption peak around 650 nm for FePcS–PMA–LDH was larger than that for FePcS–LDH.

### 3.2. Catalytic Performance of Samples

Figure 7 shows the degradation and mineralization of BPA in aqueous solution by various systems, in which the experimental conditions of the photo-Fenton reaction were decided according the influence factor experiments (Appendix A). As shown in Figure 7, negligible degradation and mineralization of BPA in the system with only visible light (curve a) indicates that BPA is quite stable under visible light irradiation. Under the dark conditions (curves b–e), the degradation and mineralization rates of BPA are rarely increased, which may be caused by the fact that H_2_O_2_ cannot be converted into •OH radicals without light irradiation. Additionally, only 28.6% of BPA was removed in the “Vis/FePcS–PMA–LDH” system (curve f), demonstrating that it is inefficient to degrade BPA when FePcS–PMA–LDH is only used as a photocatalyst. While in the “Vis/H_2_O_2_” (curve g) and “Vis/H_2_O_2_/ZnAl-LDH” (curve h) systems, the degradation efficiencies of BPA were 90.8% and 91.2% after only 80 min irradiation (Figure 7a), however, both of them had limited Total Organic Carbon (TOC) removal efficiencies, which were 18.5% and 18.9%, respectively (Figure 7b). These results indicate that the hydroxyl radicals released in this system can only degrade BPA into longer-lived intermediates. As for the “Vis/H_2_O_2_/FePcS-LDH” (curve i) and the “Vis/H_2_O_2_/FePcS–PMA–LDH” (curve j) systems, the BPA degradation achieved 100% due to the photo-Fenton like reaction [45,46,47]. However, the BPA mineralization efficiency in the Vis/H_2_O_2_/FePcS–PMA–LDH system (69.2%) within 180 min was much larger than that in the Vis/H_2_O_2_/FePcS–LDH system (41.4%). Moreover, compared with the kinetic rate constants of ZnAl–LDH, FePcS–LDH, and FePcS–PMA–LDH (shown in Figure 8), the kinetic rate constant of FePcS–PMA–LDH (0.061 min^−1^) was higher than that of ZnAl–LDH (0.022 min^−1^) and FePcS–LDH (0.032 min^−1^). These results show that FePcS–PMA–LDH has better catalytic activity than ZnAl–LDH and FePcS–LDH for the degradation of BPA under circumneutral pH condition.

The stability is also an important index in the practical application of the catalyst. The stability of FePcS–PMA–LDH was evaluated by recycling tests (Figure 9), where the results show that the degradation and mineralization reached 97.6% and 65.3% after three cycles, implying that the catalyst is relatively stable in photo-Fenton catalytic applications.

### 3.3. Photo-Fenton Mechanism Discussion

To explore the catalytic mechanism of the photo-Fenton system with FePcS–PMA–LDH, active species capture experiments were carried out. In these experiments, isopropanol (IPA), p-benzoquinone (PBQ), ammonium oxalate (AO), and sodium azide (NaN_3_) were applied as the scavengers of •OH, O_2_^•−^, h^+^ and ^1^O_2_, respectively. The results of the experiments are shown in Figure 10, it can be clearly seen that the degradation of BPA is more or less suppressed by these four scavengers. Particularly, the suppression of BPA degradation by IPA, PBQ, and NaN_3_ were more prominent than that by AO, which means that •OH, O_2_^•−^ and ^1^O_2_ play main roles in the system.

In order to understand the production path of hydroxyl radicals, superoxide radicals, and singlet oxygen, the corresponding ESR spin trapping experiments were carried out in the systems with ZnAl–LDH, PMA–LDH, FePcS–LDH, and FePcS–PMA–LDH, respectively. As shown in Figure 11, significant signals of •OH, O_2_^•−^, and ^1^O_2_ can be detected in the systems with FePcS–LDH and FePcS–PMA–LDH, which were much stronger than that in the PAM–LDH and ZnAl–LDH systems. These results demonstrate that the generation of these radicals is mainly due to the existence of FePcS. As a kind of Fe complexes, FePcS was extensively studied as a photo-Fenton catalyst, in which the iron center can react with H_2_O_2_ to form •OH [47,48]. In addition, FePcS proved that it could be excited to generate an excited state (FePcS*) upon irradiation, which can interact with O_2_ to form O_2_^•−^ and ^1^O_2_ [48]. Compared with the FePcS–LDH system, the signals of •OH, O_2_^•−^, and ^1^O_2_ were all further enhanced in the FePcS–PMA–LDH system, which verified that the addition of PMA is favorable to the formation of these active species. According to the study of Chen et al. [18], polyoxometalates (POM) can easily capture electrons from electron acceptors to form reduced POM (POM^−^). POM^−^ can not only promote the transition between Fe(III) to Fe(II) in FePc to accelerate the Fenton-like reaction, but can also directly reduce O_2_ to O_2_^•−^. Therefore, as a kind of POM, PMA has a synergistic effect with FePcS in the generation of •OH, O_2_^•−^, and ^1^O_2_.

The electrochemical measurements were carried out to investigate the process of electron transfer. The CV curves of ZnAl–LDH/GCE, FePcS–LDH/GCE, and FePcS–PMA–LDH/GCE are shown in Figure 12. The CV curves of ZnAl–LDH/GCE and FePcS–LDH/GCE show a couple of small redox peaks, whereas the FePcS–PMA–LDH modified GCE shows a pair of stable, symmetrical, and obvious redox peaks. These results imply that a faster electron transfer occurred on the surface of the FePcS–PMA–LDH/GCE electrode [49,50,51], which promotes the redox reaction between Fe(III) to Fe(II) in the iron center of FePcS.

The above results can also be confirmed by the EIS measurements. According to previous reports [52,53,54,55], the smaller radius of the Nyquist circle implies a faster electron transfer rate. As shown in Figure 13, the radius on the EIS Nyquist plot of FePcS–PMA–LDH/GCE was much smaller than that of ZnAl–LDH/GCE and FePcS–LDH/GCE, indicating a faster charge transfer occurred on the interface of the FePcS–PMA–LDH/GCE electrode. These results show that the presence of FePcS and PMA make the electron transfer easier and thus favor the photo-Fenton treatment in water.

According to the above experiments, the main contribution of PMA to the enhanced photo-Fenton activity of FePcS–PMA–LDH comes from the acceleration of electron transfer in the reaction system. The possible reaction mechanism can be proposed in Figure 14. FePcS can first be excited under the irradiation of visible light. Then, PMA can capture electrons from the excited FePcS to generate reduced PMA (PMA^−^). On one hand, PMA^−^ can reduce Fe(III)–PcS to Fe(II)–PcS, which can react with H_2_O_2_ to form •OH. On the other hand, PMA^−^ can directly act with O_2_, leading to the formation of O_2_^•−^. Furthermore, O_2_ can directly react with excited FePcS to form O_2_^•−^, and ^1^O_2_. Therefore, •OH, O_2_^•−^, and ^1^O_2_ become the main active species for BPA degradation in this system. Of course, the roles of PMA in the studied catalyst need to be further investigated.

## 4. Conclusions

A FePcS–PMA–LDH composite was successfully synthesized by the anion exchange method. The photo-Fenton catalytic activity of the as-prepared FePcS–PMA–LDH composite in the degradation of PBA was investigated under visible-light irradiation and circumneutral pH. The results showed that the FePcS–PMA–LDH composite exhibited superior catalytic activity and excellent stability. In addition, the possible photo-Fenton catalytic mechanism of FePcS–PMA–LDH was also proposed.

## Figures and Tables

**Figure 1 materials-13-01951-f001:**
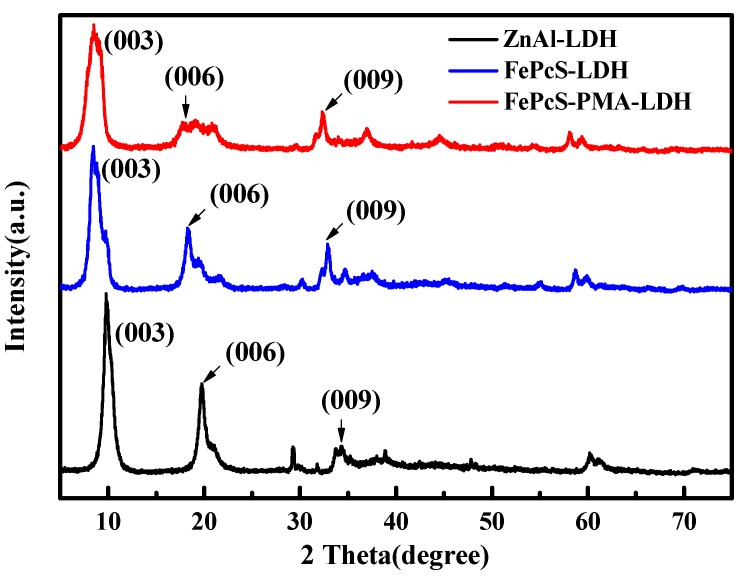
X-ray diffraction (XRD) patterns of ZnAl–LDH, FePcS–LDH, and FePcS–PMA–LDH.

**Figure 2 materials-13-01951-f002:**
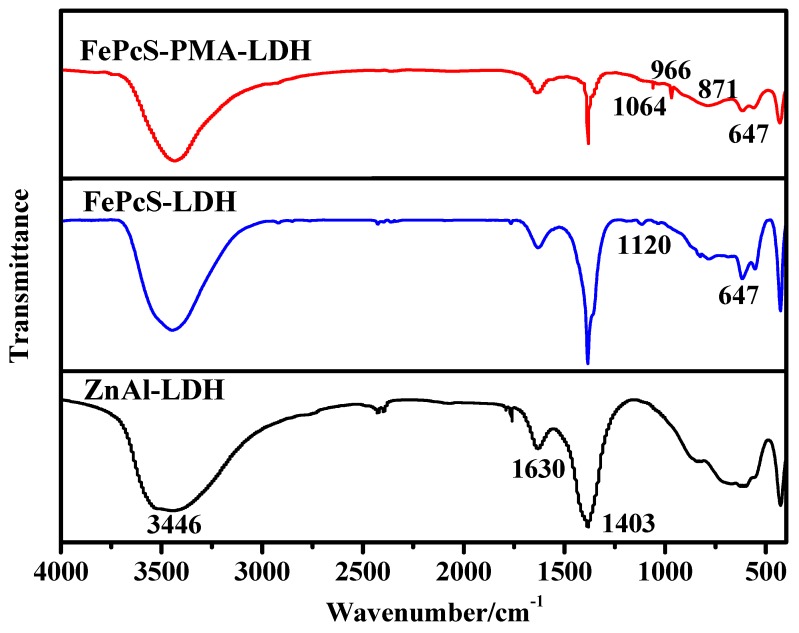
Fourier transform infrared (FTIR) spectra of ZnAl–LDH, FePcS–LDH, and FePcS–PMA–LDH.

**Figure 3 materials-13-01951-f003:**
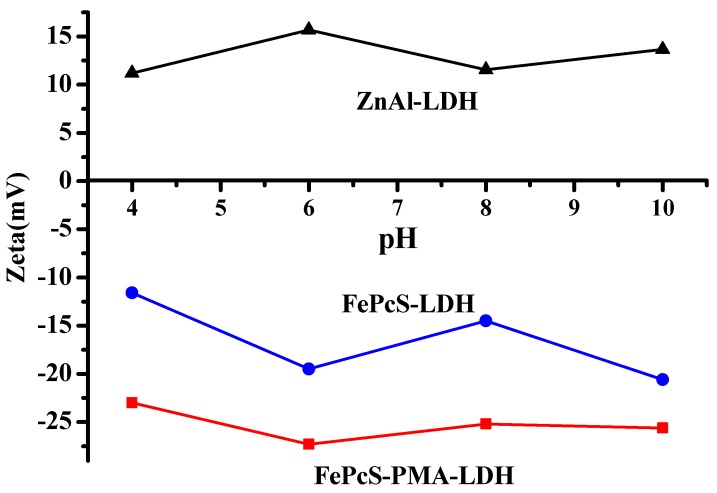
Variation of zeta potential with pH of ZnAl–LDH, FePcS–LDH, and FePcS–PMA–LDH.

**Figure 4 materials-13-01951-f004:**
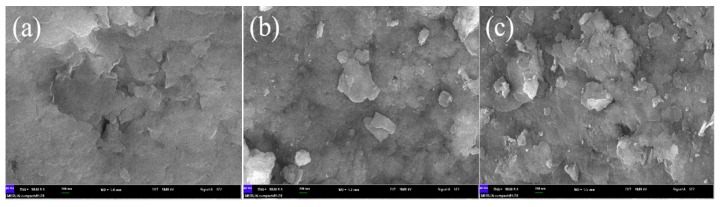
Scanning electron microscopy (SEM) images of ZnAl–LDH (**a**), FePcS–LDH (**b**), and FePcS–PMA–LDH (**c**).

**Figure 5 materials-13-01951-f005:**
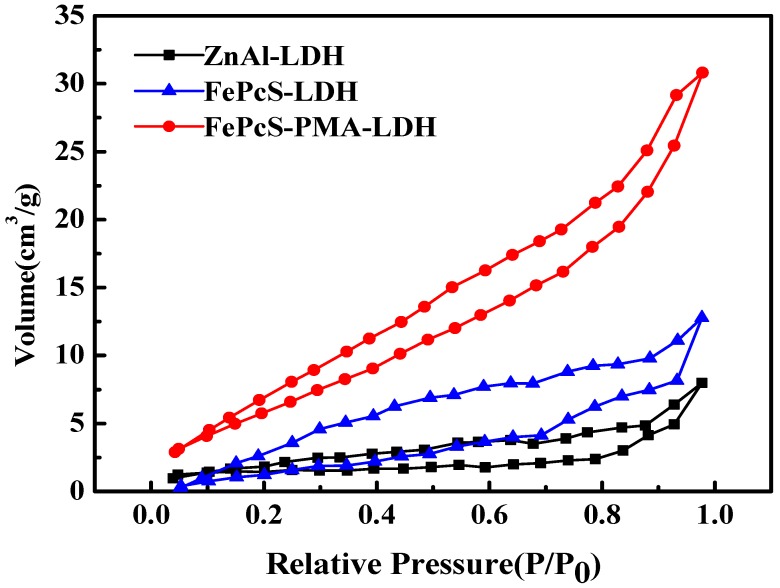
Nitrogen adsorption–desorption isotherms of ZnAl–LDH, FePcS–LDH, and FePcS–PMA–LDH.

**Figure 6 materials-13-01951-f006:**
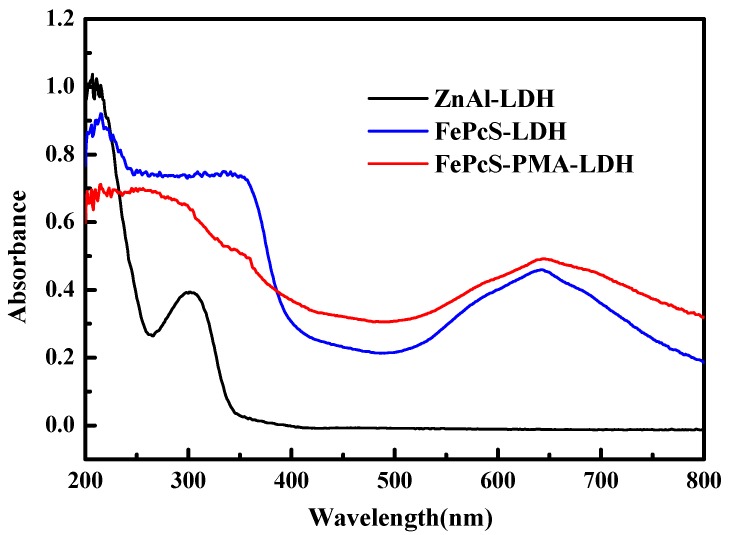
Diffuse reflectance spectra of ZnAl–LDH, FePcS–LDH, and FePcS–PMA–LDH.

**Figure 7 materials-13-01951-f007:**
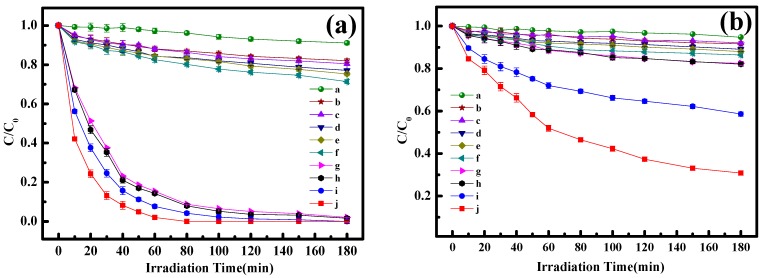
(**a**) Degradation of bisphenol A (BPA) under different conditions. (**b**) Mineralization of BPA under different conditions. (a) Vis; (b) H_2_O_2_; (c) H_2_O_2_/ZnAl–LDH; (d) H_2_O_2_/FePcS–LDH; (e) H_2_O_2_/FePcS–PMA–LDH; (f) Vis/FePcS–PMA–LDH; (g) Vis/H_2_O_2_; (h) Vis/H_2_O_2_/ZnAl–LDH; (i) Vis/H_2_O_2_/FePcS–LDH; (j) Vis/H_2_O_2_/FePcS–PMA–LDH. [BPA] = 10 mg/L; [H_2_O_2_] = 6 mM; pH = 6.0; catalyst dosage = 0.4 g/L; Light intensity = 500 W.

**Figure 8 materials-13-01951-f008:**
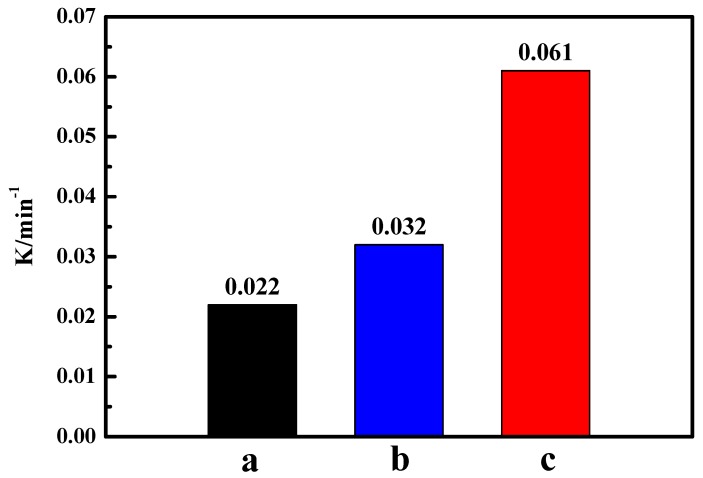
The kinetic rate constant of BPA degradation in the photo-Fenton system with different catalysts. (**a**–**c**) represent ZnAl–LDH, FePcS–LDH, and FePcS–PMA–LDH, respectively.

**Figure 9 materials-13-01951-f009:**
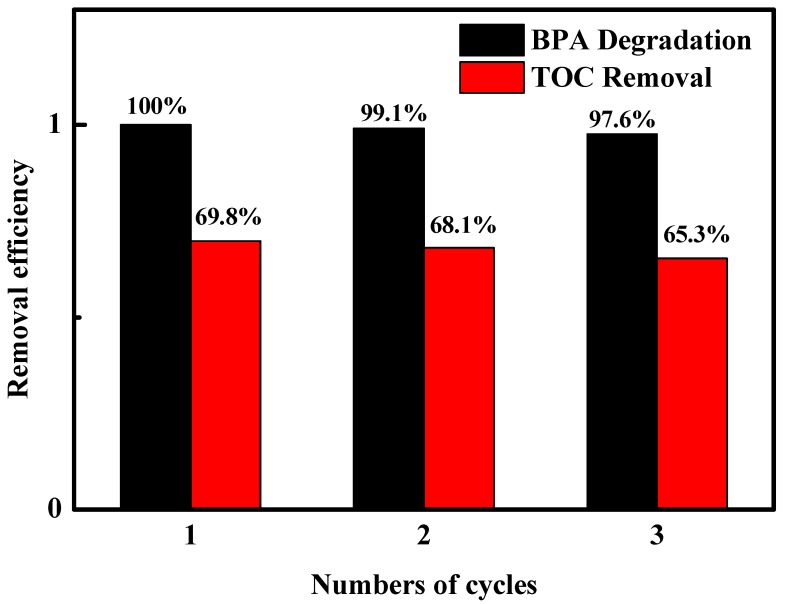
Cycling runs in the heterogeneous photo-Fenton system with FePcS–PMA–LDH.

**Figure 10 materials-13-01951-f010:**
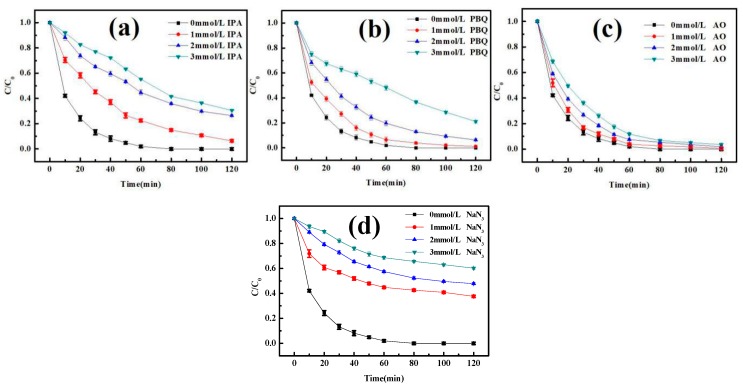
Effects of (**a**) isopropanol, (**b**) p-benzoquinone, (**c**) ammonium oxalate (AO), and (**d**) sodium azide (NaN_3_) on the degradation of BPA in the heterogeneous photo-Fenton system with FePcS–PMA–LDH.

**Figure 11 materials-13-01951-f011:**
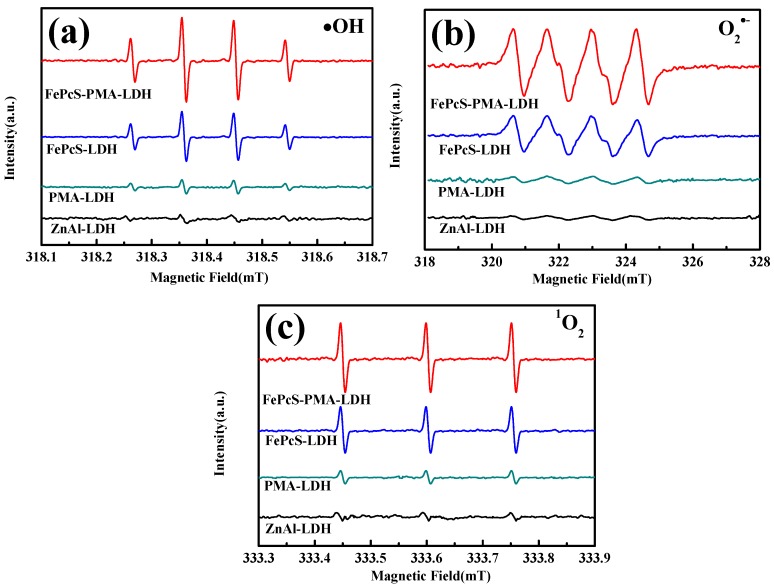
Electron spin resonance (ESR) spectra of (**a**) DMPO–HO· adducts, (**b**) DMPO–O_2_^•−^ adducts and (**c**) TEMP–^1^O_2_ adducts in the photo-Fenton system with ZnAl–LDH, PMA–LDH, FePcS–LDH, and FePcS–PMA–LDH under visible light irradiation.

**Figure 12 materials-13-01951-f012:**
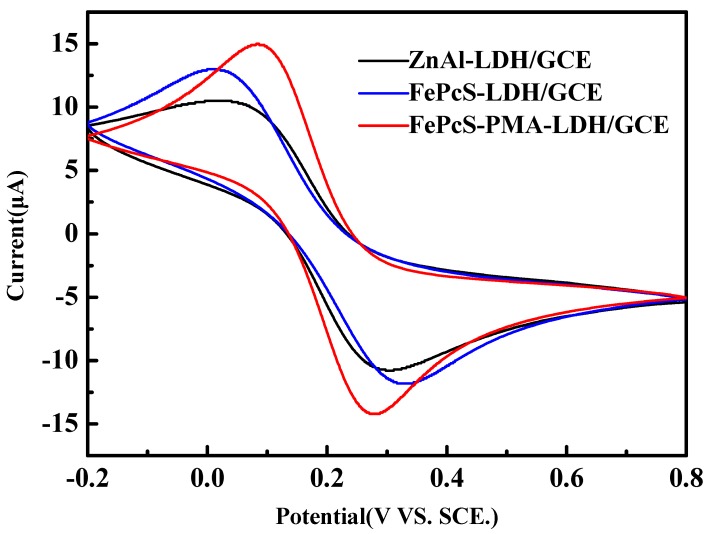
Cyclic voltammetry of ZnAl-LDH/GCE, FePcS-LDH/GCE, and FePcS–PMA–LDH/GCE in 0.1 M KCl solution containing K_3_[Fe(CN)_6_]/K_4_[Fe(CN)_6_] (both 5 mM) with a scan rate of l0 mV/s.

**Figure 13 materials-13-01951-f013:**
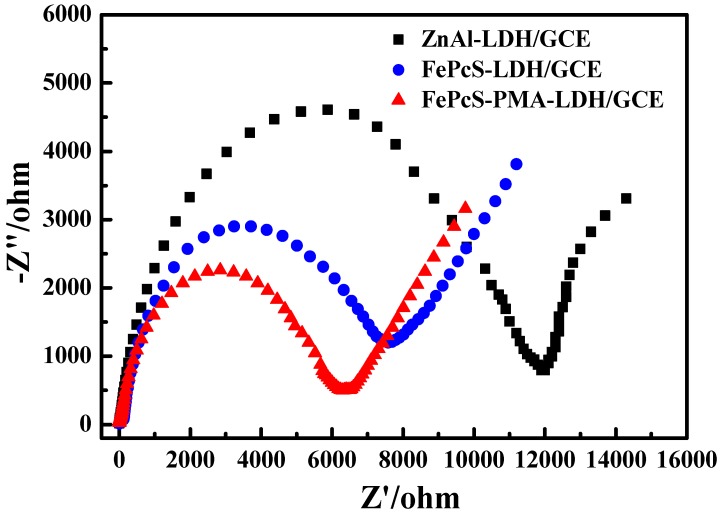
Nyquist diagrams obtained at ZnAl–LDH/GCE, FePcS–LDH/GCE, and FePcS–PMA–LDH/GCE in 0.1 M KCl solution containing K_3_[Fe(CN)_6_]/K_4_[Fe(CN)_6_] (both 5 mM).

**Figure 14 materials-13-01951-f014:**
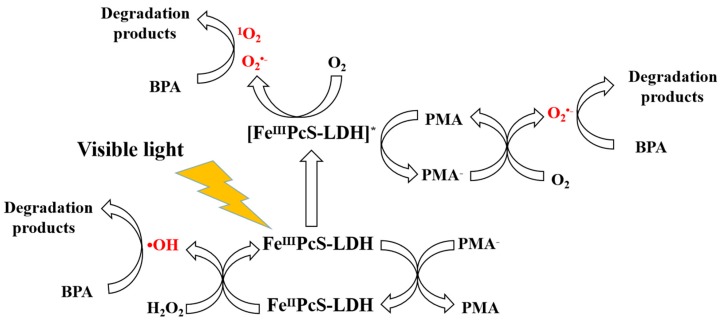
The possible photo-Fenton catalytic mechanism of FePcS–PMA–LDH to degrade BPA.

**Table 1 materials-13-01951-t001:** Basal spacing of ZnAl–LDH, FePcS–LDH, and FePcS–PMA–LDH, determined by the Bragg’s Law.

Samples	2θ (°)			Basal Spacing (nm) (003)
	(003)	(006)	(009)	
ZnAl–LDH	9.88	19.68	33.78	0.894
FePcS–LDH	8.52	18.26	32.22	1.037
FePcS–PMA–LDH	8.44	18.08	32.36	1.046

**Table 2 materials-13-01951-t002:** Brunauer–Emmett–Teller (BET) surface areas, pore volumes, and pore sizes of ZnAl–LDH, FePcS–LDH, and FePcS–PMA–LDH.

Samples	BET Surface Area (m^2^/g)	Pore Volume (cm^3^/g)	Pore Size (nm)
ZnAl–LDH	9	0.012	16.6
FePcS–LDH	16	0.031	18.0
FePcS–PMA–LDH	22	0.041	21.7

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
