# Peer review of "Synthesis of FePcS–PMA–LDH Cointercalation Composite with Enhanced Visible Light Photo-Fenton Catalytic Activity for BPA Degradation at Circumneutral pH"

_materials, 2020, doi:10.3390/ma13081951_

Round 1
Reviewer 1 Report
The paper is interesting, although before it is accepted for publication, it requires the following changes:
- The authors should review the grammar. There are many English mistakes that could have been avoided.
- Expressions such as "are demonstrated in Fig. 1" should be modified.
- Expressions like "It is should be noted", should be corrected.
- In section 3.1 (line 2), the authors say "As shown in the spectrum of ZnAl-LDH". Figure 1 does not correspond to any spectrum, but to diffractograms.
- The authors do not show any image of the composites. It would be more convenient to know how these systems are (if they are nano or microparticles, or if they correspond to another arrangement of the material). I recommend that the authors include images of electron microscopy, even if it is low resolution.
- Reflection (009) (Figure 1) is not clear. Please indicate to which peak it corresponds.
- Throughout the text there are missing superscripts (for example cm-1) and subscripts (for example H2O) that must be correctly written.
- Figures 1 and 2 must follow the same color criteria. For example, in Fig. 1 the FePcS-LDH diffractogram is red and, for the same compound, the FTIR (Fig. 2) is blue.
In addition to the comments and suggestions above, I wonder why the authors use HNO3 to adjust the pH. It is normal to use HCl, since the presence of nitrates could have an effect on catalytic activity.
On the other hand, have the authors conducted a study on the optimal catalyst concentration and pH? If this study has not been carried out, the authors must indicate on what basis they have chosen the values used in this research.
Author Response
Dear editor and reviewers:
Thank you for giving us the opportunity to revise our article entitled "Synthesis of FePcS-PMA-LDH cointercalation composite with enhanced visible light photo-Fenton catalytic activity for BPA degradation at circumneutral pH". The relevant regulations had been made in the original manuscript according to the comments of reviewers, and the major revised portions were marked in yellow bold. Now, I will reply the questions point by point:
Responses to Review #1
Reviewer #1: The paper is interesting, although before it is accepted for publication, it requires the following changes:
- The authors should review the grammar. There are many English mistakes that could have been avoided.
- i. Expressions such as "are demonstrated in Fig. 1" should be modified.
- ii. Expressions like "It is should be noted", should be corrected.
iii. In section 3.1 (line 2), the authors say "As shown in the spectrum of ZnAl-LDH". Figure 1 does not correspond to any spectrum, but to diffractograms.
- iv. Reflection (009) (Figure 1) is not clear. Please indicate to which peak it corresponds.
- v. Throughout the text there are missing superscripts (for example cm-1) and subscripts (for example H2O) that must be correctly written.
- vi. Figures 1 and 2 must follow the same color criteria. For example, in Fig. 1 the FePcS-LDH diffractogram is red and, for the same compound, the FTIR (Fig. 2) is blue.
Reply: Thanks for your careful reading. We have corrected all the grammar problems and English mistakes you mentioned above. And we have checked all the grammars of the manuscript. Besides, the reflections of (006) and (009) was added in Fig.1 and the colors of this figure were modified in the revised manuscript.
- The authors do not show any image of the composites. It would be more convenient to know how these systems are (if they are nano or microparticles, or if they correspond to another arrangement of the material). I recommend that the authors include images of electron microscopy, even if it is low resolution.
Reply: Thanks for your advice. The SEM images of ZnAl-LDH, FePcS-LDH and FePcS-PMA-LDH were investigated. As shown in Fig.R2, the surface of ZnAl-LDH is smooth. While relatively rough surfaces of FePcS-LDH and FePcS-PMA-LDH particles can be observed, which may result from the homogeneous distributions of FePcS and FePcS-PMA on the LDHs surface. It also should be noted that FePcS-PMA-LDH exhibits a more rough surface morphology compared with FePcS-LDH, indicating that FePcS-PMA-LDH can provide more adsorption sites than FePcS-LDH. Above descriptions appear on Lines 141-148, Page 5 and Fig. R1 appears as Fig. 4, in the revised manuscript.
- In addition to the comments and suggestions above, I wonder why the authors use HNO3 to adjust the pH. It is normal to use HCl, since the presence of nitrates could have an effect on catalytic activity.
Reply: Thanks. nitric acid, but not hydrochloric acid, was used in the present study for the following two reasons: Firstly, nitrates (Zn(NO3)2 and Al(NO3)3) are the raw materials of ZnAl-LDH. Thus, the use of nitric acid will not introduce new impurities into the reaction system. Secondly, hydrochloric acid contains chloride ion, which can inhibit the Fenton reaction through the following path [1,2]:
Cl-+•OH→[ClOH]• -
- On the other hand, have the authors conducted a study on the optimal catalyst concentration and pH? If this study has not been carried out, the authors must indicate on what basis they have chosen the values used in this research.
Reply: Thanks for your insightful suggestions. The optimal conditions of this research were obtained according the influence factor experiments, which were added in the supplementary materials.
References:
[1]. J.Bacardit, J.Stotzner, E.Chamarro, S.Esplugas. Effect of Salinity on the Photo-Fenton Process. Industrial & Engineering Chemistry Research. 2007,46,7615-7619.
[2].S.W.Peng,W.J.Zhang,J.He,X.F.Yang,D.S.Wang,G.S.Zeng.Enhancement of Fenton oxidation for removing organic matter from hypersaline solution by accelerating ferric system with hydroxylamine hydrochloride and benzoquinone. Journal of Environmental Sciences.2016,41,16-23.

Reviewer 2 Report
The authors describe the synthesis of a new heterogeneous photocatalyst based on iron and phtalocyanine inserted in a cointercalated double layer hydroxide. The MS is well organized, well written and the topic is pertinent; nevertheless, there are a few aspects that have to be considered before publication.
The material is synthesized and fully characterized. However, concerns appear on the applicability of the material to the photodegradation of bisphenol A compared to several controls.
-According to the authors (lines 163 and 164) “Therefore, the loading of FePcS and PMA can enhance the light absorption intensity of ZnAl-LDH in the whole light region.“ However, enhancing the absorbance in the 500-800 region is expected: delete that sentence.
-In Figure 6a the authors plot the results of the degradation of bisphenol A under several conditions. However, there are a few controls missing: i) visible light without any catalyst; visible light and FePcS-PMA-LDH (without H2O2).
-If one looks at the graph (Fig 6 and Fig 7) it seems that the K for the treatment “e” gives ca. 0.022 min-1. The authors should discuss if the synthesis and introduction of a heterogeneous catalyst with that amount of organic material (Pc) is meaningful.
-Regarding the mechanism discussion, the authors add a series of scavengers and attribute their participation only in one single kind of reaction and therefore they postulate the mechanism. This is not new in the literature but gives rise to false negative results because of the assumption of a single kind of reactivity of those scavengers. The authors should find at least additional methods to correlate their results: identification of photoproducts or/and identification of the formation of these reactive species or others.
-What is the meaning of the consideration of a scavenger of h+?
-From the electrochemical measurements the authors only discuss on the electron transfer between Fe(III) and Fe(II), is there any option for the Pc to act as an oxidant directly to the bisphenol A by electron transfer?
-In the mechanistic proposal the authors attribute the role of phosphomolybdic acid as: “On the another hand, PMA- can directly act with O2 leading to the formation of O2•-.” Can they suppot that PMA- is able to generate O2.-?
-The authors should support the participation of PMA as they postulate in the figure 12.
-Why BPA cannot capture electron from excited FePcS? or the opposite? Have the authors evaluated the redox potentials of all the species involved together with the energy of the excited state of FePcS?
-Can they generate O2.- in a different experiment and check if bisphenol A reacts with it? If it is the case they should discuss together the reactivity observed
-What about formation of singlet oxygen?
What is the meaning of this paragraph (lines 237-239)?:
“Authors should discuss the results and how they can be interpreted in perspective of previous studies and of the working hypotheses. The findings and their implications should be discussed in the broadest context possible. Future research directions may also be highlighted.”
Overall the MS needs a major revision addressing all the highlighted aspects before being considered for publication
Author Response
Dear editor and reviewers:
Thank you for giving us the opportunity to revise our article entitled "Synthesis of FePcS-PMA-LDH cointercalation composite with enhanced visible light photo-Fenton catalytic activity for BPA degradation at circumneutral pH". The relevant regulations had been made in the original manuscript according to the comments of reviewers, and the major revised portions were marked in yellow bold. Now, I will reply the questions point by point:
Responses to Review #2
The authors describe the synthesis of a new heterogeneous photocatalyst based on iron and phtalocyanine inserted in a cointercalated double layer hydroxide. The MS is well organized, well written and the topic is pertinent; nevertheless, there are a few aspects that have to be considered before publication.
- The material is synthesized and fully characterized. However, concerns appear on the applicability of the material to the photodegradation of bisphenol A compared to several controls.
Reply: Thanks for your suggestion. Indeed, there are many studies focusing on the degradation of BPA. The study of Zhu.Y.P. et al.[1] showed that almost 85% BPA was photo-Fenton degraded by Ag/AgBr/Fh catalyst in 60 min, and the research of Liu Lu et al.[2] revealed that approximately 90% methylene blue could be decomposed by Nanodiamond-CuFe-LDH catalyst in 120 min. The BPA degradation efficiency reported by these studies is lower than the result in this research. Besides, a similar FePcS-LDH catalyst was used for the degradation of methylene blue[3], in which the methylene blue can be completely degrade within 180min, but the TOC removal efficiency only reached to 53.8% within 180min. Obviously, its catalytic activity is lower than that of the catalyst FePcS-PMA-LDH studied in the present paper. By contrasting these results, it is can be known that FePcS-PMA-LDH is a high-efficiency and high catalytic activity catalyst with promising potentials for the photo-Fenton degradation of refractory organic pollutants.
- According to the authors (lines 163 and 164) “Therefore, the loading of FePcS and PMA can enhance the light absorption intensity of ZnAl-LDH in the whole light region.” However, enhancing the absorbance in the 500-800 region is expected: delete that sentence.
Reply: Thanks for your advice. This sentence was deleted in the revised manuscript.
- In Figure 6a the authors plot the results of the degradation of bisphenol A under several conditions. However, there are a few controls missing: i) visible light without any catalyst; visible light and FePcS-PMA-LDH (without H2O2).
Reply: Thanks for your meaningful advice. We have added these experiment values in Fig.7.
- If one looks at the graph (Fig 6 and Fig 7) it seems that the K for the treatment “e” gives ca. 0.022 min-1. The authors should discuss if the synthesis and introduction of a heterogeneous catalyst with that amount of organic material (Pc) is meaningful.
Reply: Thanks for your valuable comments. We have seriously considered your suggestion. The introduction of Pc in the studied catalyst can not only increase the visible light absorption range but also keep iron active in neutral solutions, thus greatly enhancing the visible light Fenton catalytic activity of the catalyst at circumneutral pH
- Regarding the mechanism discussion, the authors add a series of scavengers and attribute their participation only in one single kind of reaction and therefore they postulate the mechanism. This is not new in the literature but gives rise to false negative results because of the assumption of a single kind of reactivity of those scavengers. The authors should find at least additional methods to correlate their results: identification of photoproducts or/and identification of the formation of these reactive species or others.
Reply: Thanks for your advice. According to previous research, these scavengers can only capture certain free radicals, as shown in the following formulas:
(CH3)2CHOH+•OH→(CH3)2C•OH+H2O [4]
PBQ+O2•-→PBQ•- [5]
C2O42-+h+→2CO2 [6]
Therefore, many studies have used IPA, PBQ and AO as scavenging agents for •OH, O2•- and h+, respectively. Of course, your suggestion is very meaningful and interesting , exploring additional methods to identify these reactive species will be considered in our future work.
- What is the meaning of the consideration of a scavenger of h+
Reply: Thanks for your question. In order to investigate whether FePc was acted as a semiconductor to degrade bisphenol A [7], a scavenger of h+ was considered in this system. The experimental data show that h+ was not the main active species in the bisphenol A degradation.
- From the electrochemical measurements the authors only discuss on the electron transfer between Fe(III) and Fe(II), is there any option for the Pc to act as an oxidant directly to the bisphenol A by electron transfer.
Reply: Thanks for your suggestion. According to the recycling tests (shown in Fig.9), it can be found that FePcS-PMA-LDH is relative stable in photo-Fenton catalytic applications. Thus, we assume that the main role of Pc in this catalyst is to promote the electron transport during the redox process of iron ions. As if Pc act as an oxidant to directly degrade bisphenol A, the stability of FePcS-PMA-LDH would be decreased dramatically.
- In the mechanistic proposal the authors attribute the role of phosphomolybdic acid as: “On the another hand, PMA- can directly act with O2 leading to the formation of O2•-.” Can they support that PMA- is able to generate O2.-?
Reply: Thanks for your advice. According to previous study [8], POM can capture electrons from photocatalyst and the formed POM- can transfer electron to O2 leading to the formation of O2•-.As a kind of POM, we assumed that PMA also has the function of transferring electrons between photocatalyst (FePcS) and oxygen.
- The authors should support the participation of PMA as they postulate in the figure 12.
Reply: Thanks. As seen from the electrochemical measurements, a much faster charge transfer can be found on the interface of FePcS-PMA-LDH than that on the interface of FePcS-LDH. Hence, based on the previous reference [8], we deduce that PMA plays the role of transferring electrons between the photocatalyst and oxygen in the studied system. Of course, the roles of PMA in the studied catalyst are needed to further investigated. Above descriptions appear on Lines 244-246, Page 10 of the revised manuscript.
10.Why BPA cannot capture electron from excited FePcS? or the opposite? Have the authors evaluated the redox potentials of all the species involved together with the energy of the excited state of FePcS?
Reply: Thanks for your meaningful suggestions. Because the excited FePcS has a short lifetime (picoseconds to nanoseconds) [9], it is much more easier to react with small molecules surrounding the catalyst, such as H2O2 and O2. Besides, according to the active species capture experiments, the system contains a large number of free radicals, such as •OH, O2•- and 1O2. Therefore, the free radicals were considered as playing a leading role in BPA degradation. Of course, we cannot completely rule out the roles of the excited state of FePcS in the process of BPA degradation. However, as limited by the experimental conditions, the contribution of the excited state of FePcS in BPA degradation has not been evaluated in this work.
- Can they generate O2.- in a different experiment and check if bisphenol A reacts with it? If it is the case they should discuss together the reactivity observed.
Reply: Thanks for your advice. According to the active species capture experiments, we do found that O2•- radicals play great roles in the process of BPA degradation. So, we have modified the possible photo-Fenton catalytic mechanism of FePcS-PMA-LDH shown in Figure 13.
- What about formation of singlet oxygen?
Reply: Thanks for your valuable advice. We have supplemented the experiment of capturing singlet oxygen (Fig.R2). It is found that NaN3 has a significant inhibitory effect on singlet oxygen in this system, indicating that singlet oxygen is also a main active species in the system. Correspondingly, the possible photo-Fenton catalytic mechanism is modified. The relative information is shown on Lines 210-214, Page 8-9. Fig.R2 and Fig.R3 appear as Fig.10(d) and Fig.13 in the revised manuscript.
- What is the meaning of this paragraph (lines 237-239)? Authors should discuss the results and how they can be interpreted in perspective of previous studies and of the working hypotheses. The findings and their implications should be discussed in the broadest context possible. Future research directions may also be highlighted.
Reply: Thanks for your careful reading. We have deleted that sentence.
References:
[1].Zhu.Y.P, Zhu.R.L, Yan.L.X, Fu.H.Y, Xi.Y.F, Zhou.H.J, Zhu.G.Q, Zhu.J.X, He.H.P. Visible-light Ag/AgBr/ferrihydrite catalyst with enhanced heterogeneous photo-Fenton reactivity via electron transfer from Ag/AgBr to ferrihydrite. Applied Catalysis B:Environmental.2018, 239,280–289.
[2].Liu.Lu, Li.S.J, An.Y.L, Sun.X.C, Wu.H.L, Li.J.Z, Chen.X, Li.H.D. Hybridization of Nanodiamond and CuFe-LDH as Heterogeneous Photoactivator for Visible-Light Driven Photo-Fenton Reaction: Photocatalytic Activity and Mechanism.Catalysts.2019,9,118;doi:103390.
[3].Tang.X.X, Liu.Y. Heterogeneous photo-Fenton degradation of methylene blue under visible irradiation by iron tetrasulphophthalocyanine immobilized layered double hydroxide at circumneutral pH. Dyes and Pigments.2016,134,397-408.
[4]. Huang.W.Y, Brigante.M, Wu.F, Mousty.C, Hanna.K, Mailhot.G. Assessment of the Fe(III)−EDDS Complex in Fenton-Like Processes: From the Radical Formation to the Degradation of Bisphenol A. Environmental Science Technology.2013(47),1952−1959.
[5].Maning.L.E, Kramer.M.K, Foote.C.S. Interception of O2 by benzoquinone in cyanoaromatic sensitized photooxygenations. Tetrahedron Letters.1984(25),2523–2526.
[6]. Abbas.V, Hesam.Z.M, Mohammadmehdi.A, Hyeok.C. DPV-assisted understanding of TiO2 photocatalytic decomposition of aspirin by identifying the role of produced reactive species. Applied Catalysis B: Environmental.2020(266),118646.
[7].Cai.Y.L.Rehmana.R.A.Ke.Wu.The transition behavior of FePc on Ag(1 1 0).Chemical Physics Letters.2013,582,90-94.
[8]. Ozer, R, Ferry, J. Investigation of the Photocatalytic Activity of
TiO2-Polyoxometalate Systems. Environmental Science & Technology. 2001, 35, 3242-3246.
[9].Li.J, Ma.W.H, Huang.Y.P, Tao.X, Zhao.J.C, Xu.Y.M. Oxidative degradation of organic pollutants utilizing molecular oxygen and visible light over a supported catalyst of Fe(bpy)32+ in water. Applied Catalysis B: Environmental.2004,48,17–24.

Round 2
Reviewer 2 Report
The authors have prepared a revised verison of the original MS. However, still concerns appear before considering that this MS deserves publication.
Legend of Figure 7 and comments (lines 176-191) are not consistent. See several examples in bold:
Figure 7. (a)Degradation of BPA under different conditions. a.vis; b.H2O2; c.H2O2/ZnAl-LDH; d.H2O2/FePcS-LDH; e.H2O2/FePcS-PMA-LDH; f.vis/FePcS-PMA-LDH; g. vis/H2O2; h. vis/H2O2/ZnAl-LDH; i. vis/H2O2/FePcS-LDH; j. vis/H2O2/FePcS-PMA-LDH
As shown in Fig. 7, under the dark conditions(curves a-d) a says visible? Are b-d dark?, the degradation and minerlization of BPA are rarely decreased, indicating H2O2 cannot be converted into •OH radicals without light irradiation. While in the “Vis/H2O2” (curve e light/dark? See legend) and “Vis/H2O2/ZnAl-LDH”(curve f) systems, the degradation efficiencies of BPA are 90.8% and 91.2% after only 80min irradiation (Fig. 7a), however, both of them with negligible TOC removal efficiencies, which are 18.5% and 18.9%,respectively (Fig. 7b). This results indicate that the hydroxyl radicals released in this system can only degrade BPA into longer-lived intermediates. As for“Vis/H2O2/FePcS-LDH”(curve g no consistent with legend) and “Vis/H2O2/FePcS-PMA-LDH”(curve h) systems, due to the existence of photo-Fenton reactions (what do the authors understand by photo-Fenton reaction?), the degradation and minerlization in Vis/H2O2/FePcS-PMA-LDH system achieved to 100% and 69.2% within 180min, which are faster than Vis/H2O2/FePcS-LDH system(100% and 41.4%) I cannot see any 100% mineralization in any trace in Fig 7b. Moreover, comparing with the kinetic rate constants of ZnAl-LDH, FePcS-LDH and FePcS-PMA-LDH(shown in Fig. 8), the kinetic rate constant of FePcS-PMA-LDH (0.061min-1) is higher than that of ZnAl-LDH(0.022min-1) and FePcS-LDH (0.032min-1). These results show that FePcS-PMA-LDH has better catalytic activity than ZnAl-LDH and FePcS-LDH for the degradation of BPA under circumneutral pH condition.
-The mechanism that is prostulated in Figure 13 cannot be admitted since there are too many speculations. The authors should design experiments that give rise to different kind of photoproducts and use the trazability of photoproducts as the way of supporting their postulated mechanism.
-Therefore, design experiments to produce photodegradation of BPA by hydroxyl radicals in a clean way. Analogously for superoxide anion and singlet oxygen. Then compare the photoproducts obtained with the new photocatalyst and postulate again a new mechanism.
-Have the authors compared the use of scavengers with the other photocatalysts? Experimental conditions?
Author Response
Dear editor and reviewers:
Thank you again for giving us the opportunity to revise our article entitled "Synthesis of FePcS-PMA-LDH cointercalation composite with enhanced visible light photo-Fenton catalytic activity for BPA degradation at circumneutral pH". The relevant regulations had been made in the original manuscript according to the comments of reviewers, and the major revised portions were marked in yellow bold. Now, I will reply the questions point by point:
Question 1:
-Legend of Figure 7 and comments (lines 176-191) are not consistent. See several examples in bold:
Figure 7. (a)Degradation of BPA under different conditions. a. vis; b.H2O2; c. H2O2/ZnAl-LDH; d. H2O2/FePcS-LDH; e. H2O2/FePcS-PMA-LDH; f. vis/FePcS-PMA-LDH; g. vis/ H2O2; h. vis/ H2O2/ZnAl-LDH; i. vis/ H2O2/FePcS-LDH; j. vis/ H2O2/FePcS-PMA-LDH
As shown in Fig. 7, under the dark conditions (curves a-d) a says visible? Are b-d dark? the degradation and minerlization of BPA are rarely decreased, indicating H2O2 cannot be converted into •OH radicals without light irradiation. While in the “Vis/ H2O2” (curve e light/dark? See legend) and “Vis/ H2O2/ZnAl-LDH” (curve f) systems, the degradation efficiencies of BPA are 90.8% and 91.2% after only 80min irradiation (Fig. 7a), however, both of them with negligible TOC removal efficiencies, which are 18.5% and 18.9%, respectively (Fig. 7b). This results indicate that the hydroxyl radicals released in this system can only degrade BPA into longer-lived intermediates. As for “Vis/ H2O2/FePcS-LDH” (curve g no consistent with legend) and “Vis/ H2O2/FePcS-PMA-LDH” (curve h) systems, due to the existence of photo-Fenton reactions (what do the authors understand by photo-Fenton reaction?), the degradation and minerlization in Vis/ H2O2/FePcS-PMA-LDH system achieved to 100% and 69.2% within 180min, which are faster than Vis/ H2O2/FePcS-LDH system (100% and 41.4%) I cannot see any 100% mineralization in any trace in Fig 7b. Moreover, comparing with the kinetic rate constants of ZnAl-LDH, FePcS-LDH and FePcS-PMA-LDH (shown in Fig. 8), the kinetic rate constant of FePcS-PMA-LDH (0.061min-1) is higher than that of ZnAl-LDH(0.022min-1) and FePcS-LDH (0.032min-1). These results show that FePcS-PMA-LDH has better catalytic activity than ZnAl-LDH and FePcS-LDH for the degradation of BPA under circumneutral pH condition.
Reply: Thanks for your careful reading and valuable suggestions. We are sorry for the mistake we have made. The above comments have been modified and shown as follows: Fig. 7 shows the degradation and mineralization of BPA in aqueous solution by various systems. As shown in Fig. 7, negligible degradation and mineralization of BPA in the system with only visible light (curve a) indicates that BPA is quite stable under visible light irradiation. Under the dark conditions(curves b-e), the degradation and mineralization rates of BPA are rarely increased, which may caused by the fact that H2O2 cannot be converted into •OH radicals without light irradiation. Also, only 28.6% of BPA has been removed in the “Vis/FePcS-PMA-LDH” system(curve f), demonstrating that it is inefficient to degrade BPA when FePcS-PMA-LDH is only used as a photocatalyst. While in the “Vis/H2O2” (curve g) and “Vis/H2O2/ZnAl-LDH”(curve h) systems, the degradation efficiencies of BPA are 90.8% and 91.2% after only 80min irradiation (Fig. 7a), however, both of them with limited TOC removal efficiencies, which are 18.5% and 18.9%, respectively (Fig. 7b). This results indicate that the hydroxyl radicals released in this system can only degrade BPA into longer-lived intermediates. As for the “Vis/H2O2/FePcS-LDH”(curve i) and the “Vis/H2O2/FePcS-PMA-LDH” (curve j) systems, the BPA degradation achieves to 100% owing to the photo-Fenton like reaction[45-47]. However, the BPA mineralization efficiency in the Vis/H2O2/FePcS-PMA-LDH system (69.2%) within 180min is much larger than that in the Vis/H2O2/FePcS-LDH system(41.4%). Moreover, comparing with the kinetic rate constants of ZnAl-LDH, FePcS-LDH and FePcS-PMA-LDH(shown in Fig. 8), the kinetic rate constant of FePcS-PMA-LDH (0.061min-1) is higher than that of ZnAl-LDH(0.022min-1) and FePcS-LDH (0.032min-1). These results show that FePcS-PMA-LDH has better catalytic activity than ZnAl-LDH and FePcS-LDH for the degradation of BPA under circumneutral pH condition. The relative information is shown on Lines 178-197, page 7 of the revised manuscript.
Question 2:
-The mechanism that is prostulated in Figure 13 cannot be admitted since there are too many speculations. The authors should design experiments that give rise to different kind of photoproducts and use the trazability of photoproducts as the way of supporting their postulated mechanism.
-Therefore, design experiments to produce photodegradation of BPA by hydroxyl radicals in a clean way. Analogously for superoxide anion and singlet oxygen. Then compare the photoproducts obtained with the new photocatalyst and postulate again a new mechanism.
Reply: Thanks for your meaningful suggestions. In order to understand the production path of hydroxyl radical, superoxide radical and singlet oxygen, the corresponding ESR spin trapping experiments have been carried out in the systems with ZnAl-LDH, PMA-LDH, FePcS-LDH and FePcS-PMA-LDH, respectively. As shown in Fig.R1, significant signals of •OH, O2•- and 1O2 can be been detected in the systems with FePcS-LDH and FePcS-PMA-LDH, which are much stronger than that in the PAM-LDH and ZnAl-LDH systems. This results demonstrate that the generation of these radicals is mainly owing to the existence of FePcS. As a kind of Fe complexes, FePcS was extensively studied as a photo-Fenton catalyst, in which the iron center can react with H2O2 to from •OH [3,4]. In addition, FePcS was proved can be excited to generate excited state (FePcS*) upon irradiation, which can interact with O2 to form O2•- and 1O2 [4]. While comparing with the FePcS-LDH system, the signals of •OH, O2•- and 1O2 are all further enhanced in the FePcS-PMA-LDH system, which verified the addition of PMA is favor to the formation of these active species. According to the study of Chen et al [5], polyoxometalates (POM) can easily capture electrons from electron acceptors to from reduced POM (POM-). POM- can not only promote the the transition between of Fe(III) to Fe(II) in FePc so as to accelerate the Fenton-like reaction, but also can directly reduce O2 to O2•-. Therefore, as a kind of POM, PMA has a synergistic effect with FePcS in the generations of •OH, O2•- and 1O2. The above information can support our mechanism production shown in Fig.14 (Fig.13 in the former manuscript). The above information now appears on Lines 225-245, Page 9 and Page 10 of the revised manuscript. Fig. R1 appears as Fig.11 in the revised manuscript.
Figure R1. ESR spectra of (a)DMPO-HO· adducts, (b)DMPO-O2·− adducts and (c)TEMP-1O2 adducts in the photo-Fenton system with ZnAl-LDH, PMA-LDH, FePcS-LDH and FePcS-PMA-LDH under visible light irradiation.
Question 3:
-Have the authors compared the use of scavengers with the other photocatalysts? Experimental conditions?
Reply: Thanks for your question.
Indeed, the use of scavengers have been widely reported in the field of catalysis. For instance, Tang et al [4] reported the degradation of methylene blue (MB) in a FePcS-LDH/H2O2 photo-Fenton system under visible light. IPA, PBQ and NaN3 were applied as the scavengers of •OH and O2•- and 1O2, respectively, in which the experimental conditions was MB concentration of 20mg/L, H2O2 concentration of 6.25mM and pH of 6. Another study of Tang et al [6] investigated a heterogeneous photo-Fenton catalytic system with EDTA-Fe-LDH to degrad azocarmine B(ACB) under UV light. C3H8O(IPA), CHCl3 and NaN3 were chosen as scavengers of •OH, O2•- and 1O2 under the conditions of 0.2 g/L EDTA-Fe-LDH, 9 mM H2O2, 150 mg/LACB, and pH 6. The research of Samakchi et al [7] revealed that the MoS2/MnFe2O4/H2O2 photo-Fenton system has an efficient degradation rate on Acid Blue 113 under visible light. And the active species trapping experiments were investigated using IPA, PBQ and oxalic acid (OA) as the scavengers of •OH, O2•- and h+ under the optimum experimental conditions (pH 6.3, 0.6ml H2O2, 6.59mg catalyst). Ge et al [8] reported that silica gel (SG) can act as a photocatalyst to generate reactive oxygen species (ROS) for the degradation of polychlorinated diphenyl sulfides(PCDPSs) under the simulated sunlight irradiation. Effects of various scavengers on the photodegradation of PCDPSs was explored under pH 6. Moreover, methanol, superoxide dismutase and NaN3 were applied to the scavengers of •OH and O2•- and 1O2, respectively. Also, Gao et al [9] discussed the degradation of MB by using a metalloporphyrin-based porous organic polymer (FePPOP-1) under visible light. Under the experimental conditions of MB concentration of 70mg/L, H2O2 concentration of 30mM and pH of 7, the authors investigated the effects of different scavengers such as Methanol, PBQ and TEMPO on the efficiency of MB removal. The results finally indicated that both•OH and 1O2 were dominant ROS for MB degradation in this photo-Fenton system.
The experimental conditions of the above studies are more or less similar with the conditions in our study. Thus, we think the choice of scavengers in this research is appropriate.
Reference:
[1]. Stepanow. S,Rizzini.A.L, Krull.C,Kavich.J, Cezar.J.C,Harris.F.Y, Sheverdyaeva. P.M, Moras.P,Carbone.C, Ceballos.G, Mugarza.A, Gambardella.P. Spin tuning of electron-doped metal-phthalocyanine Layers.J.Am.Chem.Soc.2014,136,5451-5459.
[2]. Ponce.I, Silva.J.F, Onate.R,Rezende.M.C, Paez.M.A,Zagal.J.H, Pavez,J. Enhance- ment of the catalytic activity of Fe phthalocyanine for the reduction of O2 anchored to Au(111) via conjugated self-assembled monolayers of aromatic thiols as compared to Cu phthalocyanine. J. Phys. Chem. C. 2012,116,15329-15341.
[3]. Zhu,Z.X, Chen.Y, Gu.Y, Wu.F, Lu.W.Y, Xu.T.F, Chen.W.X.Catalytic degradation of recalcitrant pollutants by Fenton-like process using polyacrylonitrile-supported iron (II) phthalocyanine nanofibers: Intermediates and pathway. Water. Res. 2016, 93, 296-305.
[4]. Tang.X.X, Liu.Y. Heterogeneous photo-Fenton degradation of methylene blue under visible irradiation by iron tetrasulphophthalocyanine immobilized layered double hydroxide at circumneutral pH. Dyes and Pigments.2016,134,397-408.
[5]. Chen, C.C.; Lei, P.X.; Ji, H.W.; Ma, W.H.; Zhao, J.C. Photocatalysis by Titanium Dioxide and Polyoxometalate/TiO2 Cocatalysts. Intermediates and Mechanistic Study. Environmental Science & Technology. 2004, 38, 329-337.
[6].Tang.X.X, Liu.Y,Li.S.H.Heterogeneous UV-Fenton photodegradation of azocarmine B over [FeEDTA] intercalated ZnAl-LDH at circumneutral pH.RSC Advances.2016,6,80501.
[7].Soha.Samakchi, Naz.Chaibakhsh, Zeinab. Moradi-Shoeili. Synthesis of MoS2/ MnFe2O4 nanocomposite with highly efficient catalytic performance in visible light photo-Fenton-like process.Journal of Photochemistry&Photobiology A: Chemistry. 2018,367,420–428.
[8]. Ge.J.L, Wang.X.H, Li.C.G. Photodegradation of polychlorinated diphenyl sulfides mediated by reactive oxygen species on silica gel. Chemical Engineering Journal.2019,359,1056–1064.
[9].Gao.W.Q,Tian.J,Fang.Y.S,Liu.T.T,Zhang.X.M,Xu.X.H,Zhang.X.M.Visible-light-driven photo-Fenton degradation of organic pollutants by a novel porphyrin-based porous organic polymer at neutral pH. Chemosphere.2020,234,125334.
